# Molecular Cytogenetics Reveals Mosaicism in Human Umbilical Vein Endothelial Cells

**DOI:** 10.3390/genes13061012

**Published:** 2022-06-03

**Authors:** Regina L. Binz, Rupak Pathak

**Affiliations:** Division of Radiation Health, Department of Pharmaceutical Sciences, College of Pharmacy, University of Arkansas for Medical Sciences, Little Rock, AR 72205, USA; rkbinz@uams.edu

**Keywords:** chromosomal aberrations, spectral karyotyping, fluorescence in situ hybridization, G-banding, endothelial cells

## Abstract

Primary human umbilical vein endothelial cells (HUVECs) are consistently the most reliable in vitro model system for studying the inner lining of blood and lymphatic vessels or the endothelium. Primary human cells originate from freshly isolated tissues without genetic manipulation and generally show a modal number of 46 chromosomes with no structural alterations, at least during early passages. We investigated the cytogenetic integrity of HUVECs with conventional (G-banding) and molecular cytogenetic methods (spectral karyotyping (SKY) and fluorescence in situ hybridization (FISH)). Our G-band data shows two X-chromosomes, confirming these HUVECs originate from a female donor. Notably, some cells consistently exhibit an unfamiliar banding pattern on one X chromosome toward the distal end of the long arm (Xq). Our FISH analysis confirms that approximately 50% of these HUVECs have a deletion of the Xq terminal region. SKY analysis indicates that the deleted region is apparently not integrated into any other chromosome. Finally, we demonstrated the presence of a similar Xq deletion in the daughter cell line, EA.hy926, which was generated by fusing HUVECs with A549 (a thioguanine-resistant clone of adenocarcinomic human alveolar basal epithelial cells). These findings will advance comprehension of HUVECs biology and will augment future endothelial studies.

## 1. Introduction

The tightly packed monolayer of endothelial cells, which form the inner cellular lining of the blood and lymphatic vessels, is known as endothelium. Not only does it serve as a physical barrier between blood and the surrounding tissues [1], the endothelium also regulates blood fluidity, vascular tone, inflammation, coagulation, and immunity [2,3]. Damage to the endothelium contributes to cardiovascular disease, pulmonary hypertension, pathological angiogenesis, thrombotic thrombocytopenic purpura, hepatic veno-occlusive disease, and severe sepsis [4,5,6]. Recent studies on Coronavirus disease 2019 (COVID-19) demonstrate that damage to endothelium as a result of viral infection contributes to vascular inflammation, coagulopathy, and mortality [7,8]. Considering the significance for regulating normal physiological functions as well as a variety of pathological conditions, it is crucial to advance our knowledge on endothelial cell biology. In this regard, utilization of an ex vivo primary cell culture system is highly beneficial to better understand endothelial biology. Moreover, ex vivo systems facilitate controlled manipulation of cells throughout experimentation and could prove to be a noteworthy alternative to in vivo studies.

Since their isolation in the early 1970s, human umbilical vein endothelial cells (HUVECs) have been extensively used to understand vascular endothelial function and disease [9]. HUVECs are derived from veins exclusive to the umbilical cord. This unique biological infrastructure connects a fetus to the mother’s blood supply via the placenta, suggesting HUVECs represent endothelial cells during embryonic stage. Importantly, HUVECs share multiple attributes such as: tube formation ability, angiogenesis, wound healing, similar molecular markers, and exhibit the same modal number of chromosomes observed in post-embryonic endothelial cells. As HUVEC cells are utilized for a myriad of research applications, comprehensive cytogenetic characterization of HUVEC provides a pivotal element for appropriate utilization of these cells for research.

Conventional cytogenetic characterization has shown HUVECs to have a modal number of 46 when chromosomes are harvested during early culture passages [10]. Previous studies with conventional cytogenetic techniques reveal that long-term culture of HUVECs results in spontaneous loss of chromosome 13 or partial gain of chromosome 11, suggesting that extended culture of HUVECs is prone to induce chromosomal aberrations [10]. However, characterization of HUVECs utilizing molecular cytogenetic methods during early culture passages has not been examined. In order to potentially identify cryptic chromosomal aberrations in HUVECs during earlier passages, we used conventional G-banding in conjunction with spectral karyotyping (SKY) and fluorescence in situ hybridization (FISH), as each technique, when used alone, poses limitations.

G-banding allows visualization of recognizable bands with alternating light (GC rich) versus dark (AT rich) staining intensities (individual bands are approximately ~5–10 × 10^6^ base pairs of DNA), which are unique to each chromosome. G-banding serves the primary function of identifying aneuploidy and chromosomal rearrangements. While quite useful, this technique is limited by chromosome length and resolution of banding pattern. A low band level of resolution precludes the detection of rearrangements for some preparations, while allowing other types of aberrations to be more recognizable (pericentromeric inversions).

SKY can provide vital information to augment conventional G-band analysis. This molecular technique allows hybridization with various combinations of five fluorochromes, resulting in simultaneous identification of all chromosomes. Advanced analysis assigns pseudo colors to each chromosome based on the unique spectral emission of fluorochromes. Because SKY detects changes in a chromosome-specific manner, it facilitates identification of inter-chromosomal rearrangements, but does not reveal intrachromosomal rearrangements, such as interstitial deletions, duplications, or inversions.

FISH allows a more fine-tuned analysis by identifying locus- and/or region-specific structural chromosomal aberrations, including the sub-telomeric regions. The Vysis ToTelVysion probes specifically hybridize the p and q sub-telomeres of chromosomes 1 to 12 and 16 to 20; the q sub-telomeres of the acrocentric chromosomes (13, 14, 15, 21, and 22); and the Xp/Yp and Xq/Yq pseudo-autosomal region sub-telomeres (Table 1). In addition, several of the ToTelVysion probes contain unique sequence locus-specific identifier probes or α satellite centromere enumeration probes (CEP) for identifying specific chromosomes within the mixtures (Table 1). However, ToTelVysion probes allow analysis of only 64 specific target areas in the entire genome.

In the current study, we used G-banding, SKY, and the Vysis ToTelVysion FISH probe kit. Our studies reveal a novel cytogenetic alteration in primary HUVECs. In addition, this novel aberration is present in an immortal endothelial cell line derived by combining HUVECs with human adenocarcinoma cell line, suggesting this previously unidentified aberration is perhaps a characteristic feature of HUVECs. This novel discovery has potential to have formidable or tremendous impact for endothelial biology research.

## 2. Materials and Methods

### 2.1. Cell Culture

HUVEC cells (batch #0000560571) were purchased from Lonza (Walkersville, MD, USA) and cultured in 50/50 vascular cell basal medium (Cat # PCS-100-030, ATCC, Manassas, VA, USA) supplemented with endothelial cell growth kit-BBE (Cat # PCS-100-040, ATCC) and CHANG Medium D (Cat # 99404; FujiFilm Irvine Scientific, Santa Ana, CA, USA). EA.hy926 cells (ATCC) were cultured in Dulbecco’s Modified Eagle’s Medium (ATCC) supplemented with 10% Fetal Bovine Serum (ATCC) and 1% antibiotics. Cells were grown in a humidified incubator with 5% CO_2_ at 37 °C. Cells were subcultured every 2 to 3 d after detachment with a brief 0.25% trypsin-EDTA (Gibco, MA, USA) treatment. All cytogenetic experiments with HUVECs were performed by passage number 5.

### 2.2. Metaphase Chromosome Preparation

Metaphase chromosomes were prepared as previously described [11,12]. Semi-confluent cells were treated with KaryoMAX colcemid (Gibco, Waltham, MA, USA) at a final concentration of 75 ng/mL for 20 min to inhibit spindle fiber formation, thus arresting the cells in metaphase. The mitogen-containing media was removed, the cells were thoroughly rinsed with pre-warmed PBS with no Ca++ and Mg++ (Gibco) and treated with trypsin for detachment. All contents of the flask were collected into a 15 mL centrifuge tube and then centrifuged at 1000 rpm at room temperature for 5 min. The supernatant was carefully removed, the cell pellet was gently resuspended with a transfer pipet, 2 mL of prewarmed hypotonic solution (75 mM KCl; Gibco) was added in a dropwise manner. The final volume was brought to 6 mL and the tube was incubated in a water bath at 37 °C for 15 min. Cells were then fixed with room temperature fixative (3 to 1 volume of methanol and glacial acetic acid). Cells were washed an additional two times by repeating aspiration, fresh fixative resuspension, and centrifugation. Finally, metaphase spreads were prepared by dropwise application of cell suspension to clean slides at room temperature.

### 2.3. G-Banding

G-banded chromosomes were prepared as described elsewhere [11,12]. Slides were dried overnight at 66 °C and incubated with 0.025% trypsin for 1 min, gently rinsed with Tyrode’s buffer (Sigma, St. Louis, MO, USA), and stained with Giemsa solution (Sigma) for 5 min. Karyotype integrity and band resolution of metaphase chromosomes was determined according to the International System for Human Cytogenetic Nomenclature. Images were captured with Zeiss Imazer.Z2 microscope equipped with GenASIs Case Data Manager system, version 7.2.2.40970 (ASI, Carlsbad, CA, USA) for the analysis of karyotypes. At least 30 G-banded metaphase spreads were photographed and analyzed for each cell type.

### 2.4. FISH Hybridization

The ToTelVysion kit (Vysis Inc./Abbott Molecular Laboratories, Abbott Park, IL, USA) consists of 15-probe cocktails, to demark the sub-telomere regions of all human chromosomes. Table 1 describes the probe cocktails. Briefly, before FISH hybridization, slides were dipped in 2× SSC (Sigma) at room temperature for 5 min, then dehydrated in 70%, 80%, and 100% ethanol, respectively, for 2 min each. Slides were then air dried, placed in pre-warmed denature solution (70% formamide (Millipore, Temecula, CA, USA) in 2× SSC) at 71 °C for 1 min, then immediately dehydrated in a cold ethanol series (70%, 85%, 100%) for 2 min each. The appropriate measurement for each probe mixture was individually transferred to clean microcentrifuge tubes. All probes were prewarmed to 37 °C for 5 min, then placed in a water bath preheated to 75–77 °C for 5 min for denaturation. Each denatured probe mixture was applied to respective target area of cells, protected with a coverslip and sealed with rubber cement. Slides were protected from light in a humidified chamber and allowed to hybridize overnight at 37 °C. The following day, hybridized slides were subject to the following wash series pre-warmed to 45 °C: Formamide wash (50% formamide in 2× SSC) series of three—10 min each, followed by a series of two Coplin jars containing 2× SSC—5 min each. Slides were dipped in distilled water, immediately air dried, counterstained with DAPI, and protected with a coverslip. At least 4 cells from each hybridization area were photographed and analyzed for a total of 60 cells (GenASIs software, ASI, Carlsbad, CA, USA).

### 2.5. Spectral Karyotyping

The SKY kit from Applied Spectral Imaging (ASI) was used for this study. Slides were pre-soaked in 2× SSC solution and dehydrated in 70%, 85%, and 100% ethanol for 10 min each. Following air dry the slides were placed in pre-warmed denature solution (70% formamide (Millipore, Temecula, CA, USA) in 2× SSC) at 71 °C for 1 min, and then instantly dehydrated in a cold ethanol series (70%, 85%, and 100%) for 2 min each. Meanwhile, SKY probe was pre-warmed to 37 °C for 5 min and then denatured in a water bath at 80–81 °C for 7 min. Then, 10 μL of denatured SKY probe was added to the target area of the slide, covered with a 22 × 22-mm coverslip, and carefully sealed with rubber cement. Slides were incubated in a humidified chamber for 48 h at 37 °C for probe hybridization. Hybridized slides were washed with a series pre-warmed to 45 °C: Formamide wash (50% formamide in 2× SSC) series of three—5 min each, followed by a series of two Coplin jars containing 1× SSC—5 min each. Once removed from the 1× SSC solution, the slide was allowed to drain and 100 μL of blocking reagent was applied to the target area and protected with a plastic coverslip. Meanwhile, Cy5 antibody solution (ASI) was reconstituted with filtered 4× SSC. Following a 30 min incubation at 37 °C, the coverslip was carefully removed, Cy5 antibody solution was applied to the target area, protected with a fresh plastic coverslip, and returned to 37 °C incubator. Following 1 h of hybridization, the slide was subject to wash a series (prewarmed at 45 °C) consisting of 3 jars containing 4× SSC with 0.1% Tween-20, 5 min each. Meanwhile, Cy5.5 antibody solution (ASI) was reconstituted with filtered 4× SSC. Upon completion of the wash series, the slide was briefly dipped into distilled water to remove detergent residue, and Cy5.5 antibody solution was applied to the target area and protected with a fresh plastic coverslip. The incubation and 4× SSC with 0.1% Tween-20 wash steps were repeated, as was the brief dip in distilled water. Slides were counterstained with DAPI and covered with a clean glass coverslip. SKY images were captured using an SD200 Spectracube (ASI) mounted on a Zeiss Imager.Z2 microscope. DAPI images were captured and then inverted and enhanced with SKY View software (ASI) to produce G-band–like patterns on the axis of each chromosome. At least 30 SKY images were captured under 63× magnification for spectral analysis. Spectral imaging combines fluorescence microscopy, CCD-imaging and Fourier spectroscopy to enable simultaneous visualization for the entire spectrum at all image points, thereby assigning a unique color to each chromosome.

### 2.6. Successive Rehybridization of Slides

To prepare the slide for repeat hybridization, it was soaked in 4× SSC with 1% tween solution for 10 min, briefly rinsed in distilled water and placed into a clean Coplin jar of freshly prepared 4× SSC with 1% tween solution for an additional 10 min. The slide was gently and thoroughly rinsed in distilled water to remove any traces of detergent. It was then dehydrated in 70%, 85%, and 100% ethanol for 10 min each.

## 3. Results

### 3.1. G-Band Study Indicates Two Independent HUVEC Clones

We carefully studied the HUVEC cell line using conventional G-band staining. A total of 30 metaphase cells at approximately 400–450 band level of resolution were photographed and analyzed for any consistent change in modal number and any stable rearrangements or translocations (Appendix A). Each cell showed a 46, XX karyotype. However, the distal region of one chromosome Xq in some cells revealed an ambiguous banding pattern, thus indicating the possibility of 2 independent clones in the primary HUVEC cell line based solely on the status of chromosome Xq (Figure 1).

### 3.2. Telomere Analysis Identifies a Terminal Deletion in One Homologue of Chromosome X

Telomere FISH was performed with the ToTelVysion kit in order to elucidate a potential deletion at the distal end of chromosome Xq as observed in our G-band data and also to screen for any additional cryptic translocations or deletions involving telomere regions. This kit allows detection of aberrations involving specific target areas which are beyond the scope of resolution for conventional G-band and SKY analysis. A total of 4 cells from each of the 15 ToTelVysion cocktails were photographed (60 cells total). Chromosomes 1–22 revealed normal hybridization to each of the respective telomeres and centromere regions, indicating that each chromosome was intact with no cryptic translocations involving the telomeric region. Mix 2 showed all cells with normal hybridization to both centromeres of chromosome X, however, some of the cells displayed a normal hybridization pattern for the terminal region of the long arm of X chromosome (Xq ter), while others displayed a lack of signal on Xq ter, suggesting deletion of Xq ter. These data provide evidence of 2 clones of HUVECs based solely on the status of Xq terminal region (Figure 2, Appendix A).

### 3.3. SKY Analysis Reveals No Change in Spectral Emission

In the next step, the slide which was initially hybridized with mix 1 was subject to a second hybridization with SKY in order to validate probe localization for chromosome X, and to screen for a potential translocation/insertion of X chromatin undetected by both G-banding and FISH. A total of 30 cells (Appendix A) were photographed and carefully analyzed. While inverse DAPI banding also alluded to a subtle aberration at the distal region of chromosome Xq in some cells, SKY analysis revealed apparently normal spectral emission for all chromosomes, demonstrating that the deleted Xq ter is neither translocated to any other chromosome nor is it the receptor of any other chromatin. Figure 3 shows a representative classified karyogram of a HUVEC cell following SKY analysis.

### 3.4. Repeat Hybridization Confirms Half of the Cells Carry a Deletion of Xq Terminal Region

In order to determine the percentage of each clone, the same cells were subject to a third and final hybridization using ToTelVysion Mix 2. This strategy allows the same 30 cells photographed and analyzed for SKY to be divided into clones based on the hybridization pattern of Xq terminal region. Interestingly, this hybridization revealed a 50% split for normal Xq terminal and deleted Xq terminal regions (Figure 4, Appendix A).

### 3.5. Similar Xq Deletion Is Present in Immortal Human Endothelial Cell Line

As HUVEC is one of the parent cell lines, we hypothesize that a similar aberration might be present in EA.hy926. Metaphase chromosomes were prepared as described above, cells were hybridized with ToTelVysion mixes 1 and 2 to identify Xp and Xq, respectively. Multiple metaphase spreads were photographed and analyzed. Each of the cells revealed a normal signal pattern for Xp and centromere signals, while a deletion of Xq ter was evident in all cells (Figure 5). This data suggests that the HUVEC clone carrying Xq terminal deletion was involved in generation of EA.hy926 cell line.

## 4. Discussion

Freshly isolated primary endothelial cells from a healthy individual exhibit a normal chromosome pattern during early passages as detected by conventional cytogenetic techniques. However, endothelial cells have a tendency to accrue numerical and/or structural cytogenetic abnormalities following long-term ex vivo culture [13]. Moreover, cytogenetic alterations are common in aged individuals in comparison to their younger counterparts [14]. An earlier longitudinal study of human age-related chromosomal analysis showed that in vitro aging of adult endothelial cells results in a gain of chromosome 11 (trisomy 11), and that incidence of trisomy 11 proportionately increases over the time [13]. Similarly, a study by Zhang demonstrated that long-term culture of HUVECs results in a loss of chromosome 13 and 16, and a partial gain of chromosome 11 [10]. These studies clearly suggest that aging is a major predisposition factor for increased frequency of chromosomal aberrations; however, the nature of chromosomal aberrations is dependent upon the origin of endothelial cells. However, these studies exclusively used the conventional G-banding method, which typically yields 350–550 bands per haploid set with each band representing ~5–10 × 10^6^ base pairs of DNA. Identification of aberrations below this resolution is a major limitation of conventional cytogenetic techniques. Further, systematic analysis with relative high-resolution cytogenetic techniques have not been used in identifying chromosomal aberrations in HUVECs during early passages.

Molecular cytogenetic techniques allow identification of chromosomal aberrations at the level of 100 to 400 kilobase pairs. We used conventional and molecular cytogenetic techniques (SKY and FISH) to identify chromosomal aberrations in HUVECs during as early as passage five. We, for the first time, confirmed an Xq terminal deletion in half of the HUVECs. Terminal ends of each chromosome are protected by telomeres, which are comprised of repetitive DNA sequences. Telomeres play a crucial role in maintaining chromosomal integrity. Damage to the telomere or reduction of telomere length occurs gradually as dividing cells lose some part of the telomere with each replicative cycle. Telomere shortening/loss is a major contributing factor for numerical chromosomal aberrations in various cell types, including endothelial cells [14]. Although we identified the Xq terminal deletion, we did not observe any numerical aberrations in HUVECs, nor did we observe integration of this deleted Xq region into any other chromosome (i.e., translocation). We have not investigated the functional implications of the Xq deleted clone in endothelial cells. Notably, a deletion of Xq in peripheral blood lymphocytes contributes to abnormalities of menstruation and infertility in females [15]. Others have demonstrated that Xq deletion is associated with ovarian dysfunction [16,17]. Thus, our findings have a strong clinical relevance.

As primary cells, HUVECs have a finite lifespan and limited expansion capacity. Moreover, HUVECs, like all other primary cells, require special nutrients for their optimum growth and expansion, which are absent in most markedly available basal media. Various labs have used different growth media for HUVEC culture [9,18,19,20,21]. We found mixing of endothelial growth and Chang medium D in equal volume provides optimum culture condition for HUVECs. Considering the limitations of primary endothelial cell culture, an immortal endothelial cell line, called EA.hy926, was generated by fusing HUVECs with A549 lung cancer cells [22]. Because we observed a terminal deletion of Xq in half of the HUVECs, and considering that HUVECs are one of the parental cell types of EA.hy926, we hypothesized whether EA.hy926 cells could possibly carry the same Xq terminal deletion. Surprisingly, we observed all EA.hy926 cells show a terminal deletion of Xq, suggesting that the HUVEC clone carrying the Xq terminal deletion was involved in generation of EA.hy926.

## 5. Conclusions

In conclusion, to the best of our knowledge we identified a novel mosaic Xq terminal deletion in HUVECs and confirmed this newly identified aberration is also present in an immortal endothelial cell line generated from HUVECs. These findings will augment our knowledge of the HUVEC karyotype, improve our ability to design mechanistic studies or biomedical and pharmaceutical research with HUVEC cells, and provide critical information to apply to future endothelial biology studies. In the future, we plan to identify the cytogenetic signatures of oncogenic transformation in HUVECs following long-term culture.

## Figures and Tables

**Figure 1 genes-13-01012-f001:**
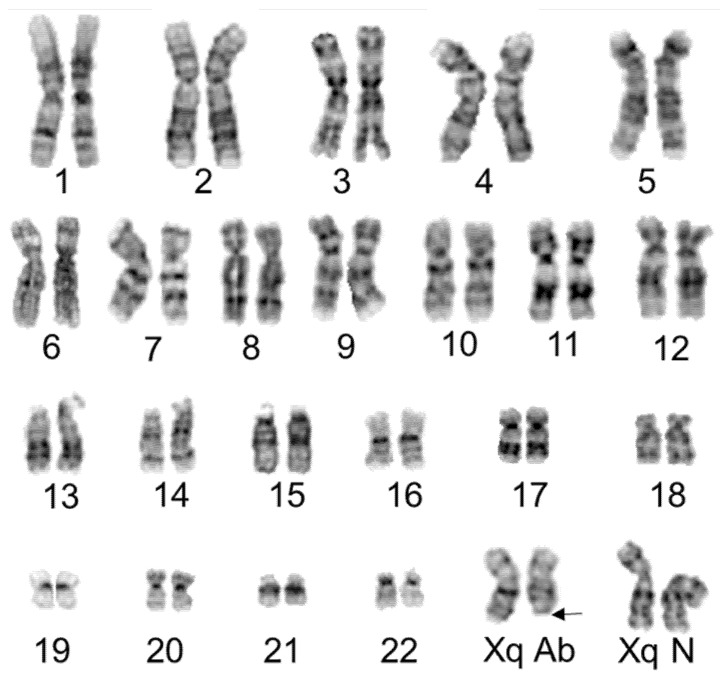
Composite karyogram of HUVEC cells at the 425-band level of resolution. Chromosomes 1–22 showed no consistent variation of banding pattern, while several cells displayed abnormal morphology in the terminal region of the long arm of chromosome X (Xq Ab as indicated by arrow) and the rest of the cells exhibit normal Xq region (Xq N). Images were captured under 100× magnification.

**Figure 2 genes-13-01012-f002:**
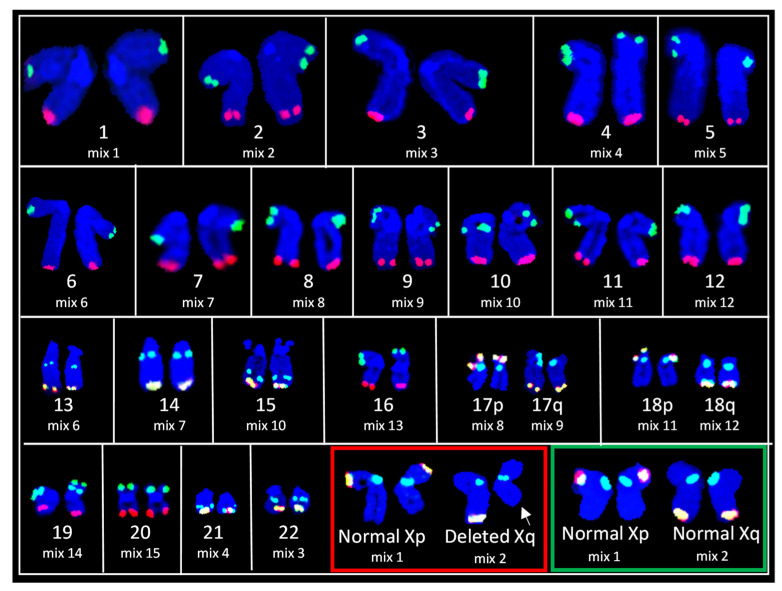
Composite karyogram from 15 metaphase spreads hybridized with the ToTelVysion (TV) subtelomeric probe set. Each chromosome pair is labeled with its respective TV probe mix (see Table 1). Red and green boxes show consecutive hybridizations for deletion of chromosome Xq terminal and normal Xq terminal regions, respectively. White arrow indicates deletion of Xq terminal region (see Appendix A for metaphase images).

**Figure 3 genes-13-01012-f003:**
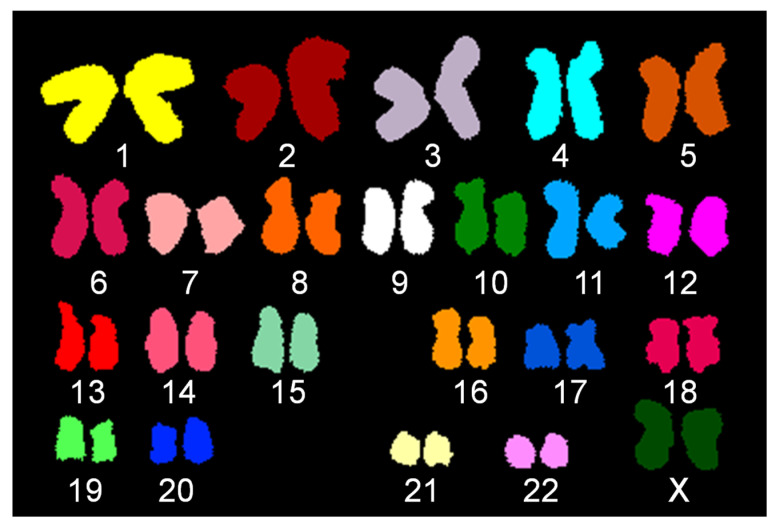
Classified image from Spectral Karyotype (SKY) of HUVECs. Classified image shows no interchromosomal translocation.

**Figure 4 genes-13-01012-f004:**
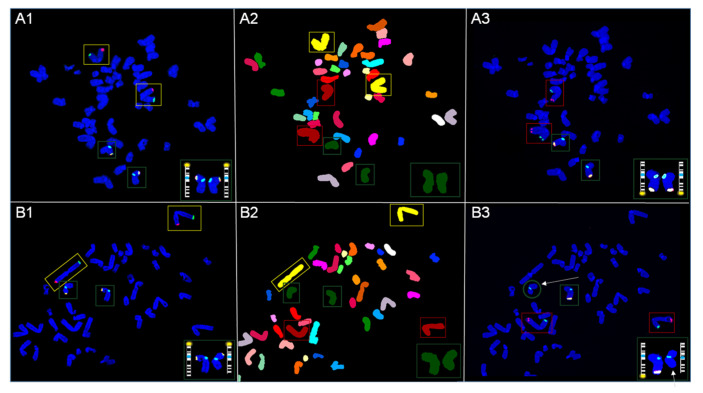
Photomicrographs showing normal Xq and abnormal Xq following 3 consecutive hybridizations of metaphase spreads. Each frame contains an insert representing the magnified image with representative ideogram of chromosome X for 3 respective hybridizations. (**A1**) First hybridization with TV mix 1 reveals a normal signal pattern for chromosomes 1 (yellow boxes) and Xp (green boxes). (**A2**) Classified image as a result of second hybridization with SKY, validates FISH hybridizations for chromosomes X (green boxes), 1 (yellow boxes), and 2 (red boxes), while illustrating no change in spectral emission for Xq ter. (**A3**) Third hybridization with TV mix 2 reveals a normal signal pattern for chromosomes 2 (red boxes) and Xq (green boxes). (**B1**) First hybridization with TV mix 1 reveals a normal signal pattern for chromosomes 1 (yellow boxes) and Xp (green boxes). (**B2**) Classified image as a result of second hybridization with SKY, validates FISH hybridizations for chromosomes X (green boxes), 1 (yellow boxes), and 2 (red boxes), while illustrating no change in spectral emission for Xq ter. (**B3**) Third hybridization with TV mix 2 reveals a normal signal pattern for chromosomes 2 (red boxes), one chromosome X with a normal signal pattern for Xq (green box) and one chromosome X with a deletion of Xq ter (green circle). White arrows indicate deletion of Xq terminal region.

**Figure 5 genes-13-01012-f005:**
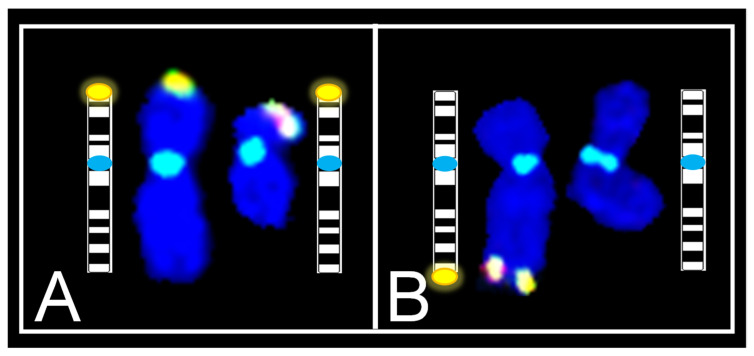
Representative photomicrographs showing a deletion of Xq terminal region in EA.hy926 cells. (**A**) ToTelvysion mix 1 shows a normal signal pattern for X CEP and Xp telomeres and (**B**) ToTelVysion mix 2 reveals a normal signal pattern for X CEP in both chromosomes and a deletion of one homologue of chromosome Xq telomere. Idiograms of chromosome X showing Xp (**A**) and Xq (**B**) terminal regions (highlighted).

**Table 1 genes-13-01012-t001:** A summary of probe loci and their respective fluorochrome colors for each of the 15 mixtures included in the ToTelVysion kit. CEP, centromere; p, short arm; and q, long arm.

ToTelVysion Probes				
Probes	Green	Orange	Yellow	Aqua
Mix 1	1p	1q	Xp	X CEP
Mix 2	2p	2q	Xq	X CEP
Mix 3	3p	3q	22q	22q11
Mix 4	4p	4q	21q	21q22
Mix 5	5p	5q	n/a	n/a
Mix 6	6p	6q	13q	13q14
Mix 7	7p	7q	14q	14q11.2
Mix 8	8p	8q	17p	17 CEP
Mix 9	9p	9q	17q	17 CEP
Mix 10	10p	10q	15q	15q22
Mix 11	11p	11q	18p	18 CEP
Mix 12	12p	12q	18q	18 CEP
Mix 13	16p	16q	n/a	n/a
Mix 14	19p	19q	n/a	19p13
Mix 15	20p	20q	n/a	n/a

## Data Availability

Not applicable.

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
