# Peer review of "Molecular Cytogenetics Reveals Mosaicism in Human Umbilical Vein Endothelial Cells"

_genes, 2022, doi:10.3390/genes13061012_

Round 1
Reviewer 1 Report
Dear Authors,
This article holds an immense scientific novelty with adequate experimental investigations and was well written. The following minor corrections are suggested:
1. Make italicized the words, "ex vivo" and "in vivo" on page one on lines 39 and 41.
2. Please write, "∼5–10 × 106 base" as, "∼5–10 × 106 bases" on page 2 on line 63.
3. Please write, "5% CO2" as "5% CO2" on page 3 on line 102.
4. Mention in Figure 1 whether this figure of karyogram was performed at 100X magnification or more magnification.
5. Please keep a gap; as Figure 1 legends were covered by Figure 2.
Thanks.
Author Response
This article holds an immense scientific novelty with adequate experimental investigations and was well written. The following minor corrections are suggested:
- Make italicized the words, "ex vivo" and "in vivo" on page one on lines 39 and 41.
Response: Necessary corrections have been made.
- Please write, "∼5–10 × 106 base" as, "∼5–10 × 106 bases" on page 2 on line 63.
Response: Corrected
- Please write, "5% CO2" as "5% CO2" on page 3 on line 102.
Response: Corrected
- Mention in Figure 1 whether this figure of karyogram was performed at 100X magnification or more magnification.
Response: The magnification of karyogram image has been incorporated in the figure legend.
- Please keep a gap; as Figure 1 legends were covered by Figure 2.
Response: We apologize for this mistake. Necessary corrective measures have been taken.

Reviewer 2 Report
The manuscript entitled “Molecular cytogenetics reveals mosaicism in human umbilical 2 vein endothelial cells” is focussed on cytogenetic characterization of human umbilical vein endothelial cells (HUVECs), which the authors has procured from Lonza (Walkersville, MD) and have reported a deletion of the Xq terminal region through classical G-banding, Spectral Karyotyping and FISH. Authors also compared this detected deletion with a transformed cell line n EA.hy926.
The entire study is comprehensive, and well organized and the result is novel. This reviewer opines that the report of deletion from the terminal part of Xq is important as this finding has characterized the given cell line as a genetic mosaic. This report will influence the future workers with this cell line considering it is genetically heterogeneous. Secondly, finding out the similarity with the transform cell line is also significant to anticipate if the current cell line would change into cancer cell line with the passage of time.
This reviewer suggests publication of the manuscript with a round of minor revisions. The following points are to be addressed.
1. The authors must state one or two lines in the INTRODUCTION regarding their choice of HUVEC cells (batch #0000560571) over other cell lines and cite previous studies that used this cell line, if possible.
2. On page 2, lines 63-64, the authors stated “G-banding allows visualization of recognizable bands with alternating light (GC 62 rich) versus dark (AT-rich) staining intensities (each band approximately ∼5–10 × 106 bases”. Is this value correct? Or it would be ∼5–10 × 106 bases? Please check. There are other places where the value is presented in the same manner.
3. At DISCUSSION the authors must add a few lines explaining the probable origin of this terminal deletion in HUVEC cells (batch #0000560571). Is it generated de novo in due course of the passage of time in culture or was it present in the original host?
4. In CONCLUSION the authors must highlight if this cell line has already started expressing the molecular feature of the transformed cell line and should mention that this would be the subject of future study.
Author Response
The manuscript entitled “Molecular cytogenetics reveals mosaicism in human umbilical 2 vein endothelial cells” is focussed on cytogenetic characterization of human umbilical vein endothelial cells (HUVECs), which the authors has procured from Lonza (Walkersville, MD) and have reported a deletion of the Xq terminal region through classical G-banding, Spectral Karyotyping and FISH. Authors also compared this detected deletion with a transformed cell line n EA.hy926.
The entire study is comprehensive, and well organized and the result is novel. This reviewer opines that the report of deletion from the terminal part of Xq is important as this finding has characterized the given cell line as a genetic mosaic. This report will influence the future workers with this cell line considering it is genetically heterogeneous. Secondly, finding out the similarity with the transform cell line is also significant to anticipate if the current cell line would change into cancer cell line with the passage of time.
This reviewer suggests publication of the manuscript with a round of minor revisions. The following points are to be addressed.
- The authors must state one or two lines in the INTRODUCTION regarding their choice of HUVEC cells (batch #0000560571) over other cell lines and cite previous studies that used this cell line, if possible.
Response: We have revised the introduction as suggested by the reviewer.
- On page 2, lines 63-64, the authors stated “G-banding allows visualization of recognizable bands with alternating light (GC 62 rich) versus dark (AT-rich) staining intensities (each band approximately ∼5–10 × 106 bases”. Is this value correct? Or it would be ∼5–10 × 106 bases? Please check. There are other places where the value is presented in the same manner.
Response: We apologize for this mistake. We corrected the base number in our revised version.
- At DISCUSSION the authors must add a few lines explaining the probable origin of this terminal deletion in HUVEC cells (batch #0000560571). Is it generated de novo in due course of the passage of time in culture or was it present in the original host?
Response: This deletion was observed at passage as early as 5. Therefore, the change of de novo generation of this deletion is minimum. We believe the deletion was present in the original host.
- In CONCLUSION the authors must highlight if this cell line has already started expressing the molecular feature of the transformed cell line and should mention that this would be the subject of future study.
Response: We have not noticed any oncogenic transformation in HUVECs in passage 5. We acknowledge that oncogenic transformation may develop during long-term HUVEC culture, which is beyond the scope of current study. We have added a sentence about our future plan in Conclusion.